# Ecological Restoration of Engineering Slopes in China—A Review

**Yifan Shen** [1], **Qi Li** [1,2,*], **Xiangjun Pei** [1,2,*], **Renjie Wei** [1], **Bingmei Yang** [3], **Ningfei Lei** [1,2], **Xiaochao Zhang** [1,2], **Daqiu Yin** [4], **Shijun Wang** [4] and **Qizhong Tao** [4]

1   College of Ecological and Environment, Chengdu University of Technology, Chengdu 610059, China
2   State Key Laboratory of Geohazard Prevention and Geoenvironment Protection, Chengdu University of Technology, Chengdu 610059, China
3   College of Environment Science and Engineering, Guilin University of Technology, Guilin 541004, China
4   China Huaneng Group Co., Ltd., Beijing 100031, China
*   Correspondence: liqi21@cdut.edu.cn (Q.L.); peixj0119@tom.com (X.P.); Tel.: +86-15397732365 (Q.L.); +86-13550167465 (X.P.)

**Abstract:** As the protection of the environment gains more public attention in China, a large number of engineering slopes, which are not conducive to the growth of vegetation and are prone to natural disasters caused by constructions, are in urgent need of restoration. Herein, we explain the theoretical basis for the ecological restoration of engineering slopes and introduce the technologies commonly used in this regard, including soil improvement, bioremediation, and ecological slope protection. The benefits and evaluation of the impact of ecological restoration of engineering slopes are also detailed. Finally, we discuss the current problems in ecological restoration and put forward some future research prospects. By summarizing the existing techniques and evaluation systems for ecological restoration, this study provides a reference for its implementation and evaluation, contributing to the long-term, stable, and rapid development of ecological restoration of engineering slopes.

**Keywords:** engineering slopes; ecological protection; ecological restoration; evaluation

## 1. Introduction

With the rapid development of China's social economy, a large number of roads, railways, hydropower stations, and the construction of other infrastructure are being carried out. Meanwhile, the construction efforts inevitably destroy the natural environment, creating a large number of engineering slopes [1]. These not only lead to natural disasters, such as landslides, but may also affect the safety and efficiency of construction. Moreover, due to the excavation process, a large quantity of seeds may transfer from the topsoil, resulting in the decline of biodiversity, unbalancing the ecological environment and adversely affecting the sustainable development of the regional economy [2–4]. Therefore, the ecological restoration of engineering slopes is not only of great significance to protect "the lucid waters and lush mountains" but also promotes sustainable national economic growth [5].

Nowadays, China has been paying increasing attention to the ecological environment. Many studies focusing on the ecological restoration of engineering slopes and ecological restoration technologies have been carried out [6–9]. Since 2000, the area of vegetation under restoration following slope engineering has increased at the rate of 200~300 million m$^2$ per year [10–12]. Even so, the concept of ecological restoration in slope engineering is often ignored due to the delayed research on ecological restoration technologies. Many practical challenges still remain to be resolved in the field of ecological restoration, and a large number of new engineering slopes are constantly being created along with the new constructions. Therefore, timely and effective ecological restoration

of engineering slopes is necessary. This review discusses why we need to restore engineering slopes and how to realize it. To obtain related literatures, the authors searched in all databases on the Web of Science and CNKI with the keywords ecological restoration of engineering slopes, biochar, microbial agents, bioremediation, ecological slope protection, and effect evaluation of ecological restoration, separately. Among them, the authors summarize some commonly used and emerging ecological restoration technologies. The existing problems are discussed and some expectations for its further advancement are put forward. This review aims to provide a reference for the implementation of high-quality ecological restoration and its effective evaluation.

## 2. Theoretical Basis of Engineering Slopes

Damaged ecosystems cannot recover through self-recovery under natural conditions. Therefore, methods of ecological restoration that are based on human intervention are helpful in restoring damaged ecosystems. The ecological restoration of engineering slopes refers to the process of accelerating the recurrence of the original vegetation community, or the reestablishment of a new vegetation community. Appropriate artificial measures were carried out on the restoration of different construction sites, which were caused by the geomorphology of digging, collapse, and stacking. Finally, ecological functions can be restored through appropriate methods. As a result, ecological restoration can conserve soil and water, regulate microclimate, keep the biodiversity and economic functions, and develop and utilize ecosystem [13,14].

A reasonable and clear objective is an indispensable part of a successful ecological restoration plan. Often, the goal is to return the polluted or damaged ecosystems to its original function, or to restore the damaged environment to a baseline or acceptable risk level [15]. The restoration plan is often set according to the three states of degradation: undegraded, partially degraded, and highly degraded state [16]. Additionally, the goal of ecological restoration is to ultimately and particularly form a stable soil vegetation system, creating a "consolidation-maintenance" relationship between the soil and the vegetation within the ecosystem. The vegetation stabilizes the soil through the soil-reinforcement by deep roots, whereas the soil nourishes the vegetation. Finally, the ecological restoration can not only reduce soil erosion, rock weathering, landslides and other regional geological disasters, but can also restore the function of biodiversity and ecosystem [17,18].

## 3. Study on the Ecological Restoration Technologies of Engineering Slopes

The research of ecological restoration of engineering slopes in developed countries started earlier than in China. In the 1940s, the technology of vegetation slope protection was carried out to restore highway slopes in the United States [14]. Since the 1960s, environmentally friendly methods of road construction were widely accepted to minimize the occurrence of engineering slopes in the Netherlands [19]. Meanwhile, a variety of technologies have been developed and applied, such as hydraulic spray seeding technology, vegetation concrete, three-dimensional net and grass planting slope protection technology, or fiber soil greening in Japan [14].

In China, the subject of ecological restoration of engineering slopes has gained increased attention since the 1990s. With advancements in the technology sector, huge achievements were made in soil improvement, bioremediation, and ecological slope protection.

### 3.1. Method of Soil Improvement

Soil is the medium for plant growth, the basis for agricultural production, and the natural filtration system for surface water. Therefore, it is an indispensable part of the ecological environment and plays an extremely important role in the survival and development of human beings [20]. The stripping and compaction of topsoil by construction often leads to soil degradation and erosion, resulting in desertification, decline in agricultural productivity and soil ecosystem functions [21]. Therefore, the restoration of soil damaged by construction is of the utmost importance.

The use of organic amendments is an effective to induce soil improvement. A rational use of organic amendments can increase the content of soil organic matter, improve soil fertility and diversity, and purify the polluted soil [20,22]. Organic amendments, such as biochar, animal manure, or compost have been frequently used for soil improvement. According to different feedstocks, there are five categories of organic amendments: animal manure, municipal biosolids, green manure and covering crops, waste produced in the production process, and compost [23]. Biochar is a highly aromatic carbon-rich product, resulting from pyrolysis and carbonization of biomass under anoxic or anaerobic conditions [24]. This organic amendment presents a stable structure and good physicochemical adsorption capacity, which can enhance the fertility and carbon sequestration capacity of degraded soils [24–26]. Biochar feedstocks come from a wide range of sources, and the pyrolysis process can turn harmful biomass into harmless resources, thus reducing the amount of waste in the biochar manufacturing process [27,28]. Although pyrolysis in poorly built plants (such as stoves and drum kilns) may have a negative impact on greenhouse gas emissions and induce the release of highly toxic compounds, it can help minimize the emissions by re-using the pyrolytic gases and waste heat or by controlling the pyrolysis temperature within a certain range [29,30]. Therefore, biochar has a good development prospect in ecological restoration of soil. The feedstocks and manufacturing processes of different biochar are summarized in Table 1 [31–44].

**Table 1.** Feedstocks and manufacturing processes of some biochar.

| Feedstock | Manufacturing Process | Reference |
|---|---|---|
| Shredded cotton stalk | Slowly pyrolyzed at 450 °C under limited oxygen | [31] |
| Sugar maple and red maple | Pyrolyzed at 500 °C under argon atmosphere for 30 min | [33] |
| Eucalyptus saligna leaves | Pyrolyzed at 400 °C or 550 °C without steam activation | [32] |
| Paper sludge, corn stover, dried distillers grains with soluble, pinewood sawdust, cow manure, and wastewater biosolids | Pyrolyzed from 600 °C to 800 °C by catalytic pyrolysis of a catalyst made at the same temperature | [35] |
| Prune residues from orchards | Discontinuously pyrolyzed at 500 °C | [34] |
| Pinewood, peanut shell, and bamboo | Produced by slow pyrolysis at temperatures from 300 °C to 500 °C or through hydrothermal conversion | [36] |
| Eucalyptus globulus or chopped Lantana camara stem | Slowly pyrolyzed at 500 °C, and then grounded to 2 mm granules | [37] |
| Parthenium weed | Pyrolyzed from 200 °C to 500 °C for 30 to 120 min | [38] |
| O. Ficus cladodes | Pyrolyzed at 200 °C for 30 min, and then at 600 °C for an hour | [39] |
| The residual biomass of cultivated Gracilaria after agar extraction and Oedogonium | Soaked in FeCl$_3$ solution for 24 h, then dried in an oven at 60 °C for 24 h, and finally slowly pyrolyzed at 300, 450, and 750 °C for 60 min | [40] |
| Eichornia crassipes | Pyrolyzed with limited oxygen from 200 to 500 °C for 30 min to 2 h | [41] |
| Seaweed powder | Soaked in KOH solution, then dried in an oven at 80 °C, and at last pyrolyzed at 700 °C for 270 min under the atmosphere of nitrogen | [42] |
| Tea waste | Pyrolyzed at 300, 500, and 700 °C for 1 h under the condition of limited oxygen | [43] |

### 3.2. Bioremediation

The process of bioremediation in the ecological restoration of engineering slopes includes phytoremediation, microbial remediation, and plant-microbial combined remediation. Among them, the process of microbial remediation of degraded or eroded soil has been widely used in practical applications due to its low capital investment, simple operation, and no secondary pollution [45,46]. The application of microbial agents can promote the soil microflora richness, enzyme activity, plant growth, and resistance by improving the ability of roots to absorb, fix, and transform nutrients, although the effective

time of microbial agents on soil improvement still can be improved [47–50]. Phosphorus solubilizing microorganisms convert insoluble phosphorus in the soil into soluble phosphorus, which can be absorbed and utilized by plants [51]. Additionally, these microorganisms can stimulate plants to absorb trace elements such as Fe and Zn, and improve their ability to absorb nutrients from the soil [52]. In addition, the use of EM (effective microorganisms) agents can ensure plant yield and reduce the application of chemical fertilizers, which can effectively avoid soil pollution caused by the excessive use of fertilization [53]. As shown by the results presented in Table 2, microbial agents show positive effects on soil improvement [44,54–59]. Further, the combination of microbial agents and other materials is another efficient method for ecological restoration. In order to restore the degraded soil in an open-pit mining area, a combination of modified water-jet loom sludge and microbial agents was used. This not only significantly improved the water holding capacity of soil, but also reduced the pH of saline-alkali soil, and increased the content of soil organic carbon and nutrients. As a result, it promoted the growth of plants [60].

**Table 2.** Some microbial agents and their effects on soil improvement.

| Efficient Bacterium | Improvement of Soil | References |
|---|---|---|
| Bacillus megaterium | Reduce the electrical conductivity and total salt content of secondary salinized soil, increase phosphatase and urease activity, enhance the soil organic matter content, and decrease the concentration of $NO_3^-$ and $Ca^{2+}$ | [55] |
| Bacillus amyloliquefaciens | Increase the content of total salt, organic matter, and nutrients in soil; improve the utilization ability of soil microorganisms to substrate carbon sources; and improve the richness of microbial population in soil | [56] |
| Bacillus subtilis, Bacillus licheniformis, Bacillus laterosporus, Actinomycetes | Reduce the salt content and pH of topsoil, and increase the soil available nutrient and organic matter content | [57] |
| Bacillus cereus, Stretococcus thermophiles, Bacillus mucilaginosus, Bacillus subtilis, Lactobacillus plantarum, Candidautilis | Significantly increase the number of bacteria and fungi in sandy soil, improve soil nutrient level and sustainability, increase plant chlorophyll content, and enhance photosynthesis | [58] |
| Bacillus halotolerans, Sinorhizobium meliloti, Bacillus megaterium, Bacillus subtilis | Increase the soil organic matter and nutrient content, enhance sucrase and urease activity, and promote plant growth | [44,54] |
| Azotobacter salinestris, Streptomyces bacillaris, Bacillus amyloliquefaciens, Paenibacillus mucilaginosus, Bacillus subtilis | Reduce the pH, electrical conductivity, and total salt content of saline-alkali soil; increase the content of soil organic matter, alkali-hydrolyzed phosphorus and available phosphorus; bacteria in microbial agents can become the dominant bacteria in saline-alkali soil | [59] |

As an important part of bioremediation, phytoremediation has been widely used in ecological restoration of engineering slopes. The soil seed bank is the sum of all surviving seeds on the soil surface and in the soil. As a potential plant community, it plays a vital role in the restoration of vegetation [61]. Previous studies have shown that local soil seed banks can be used for restoring vegetation under certain conditions. Compared to alien plants, local soil seed banks can better adapt to the environment. Therefore, they are more conducive to seed germination and have higher economic benefits [61]. However, the selection of plant species should not only consider local soil seed banks. In fact, priority should also be given to the combined application of fast-growing and resistant plants [62]. For example, Lolium perenne L. can bind and adsorb sand particles that lack cohesion by secreting viscose during growth. The soil strength is increased through mechanical properties due to structural reinforcement by shallow roots and anchorage by deep roots [63]. Root-soil interactions, through the tensile ability of root system, can lead to a stabilization of slopes [51]. The use of vegetation for soil reinforcement and slope stabilization will not only create favorable growth conditions for subsequent plants, but also lead to a better coverage under extreme natural conditions [64].

### 3.3. Method of Ecological Slope Protection

The method of ecological slope protection consists of the combination of soil improvement and bioremediation. Through certain engineering measures, plants and some plant-like materials are combined to re-establish an ecological environment suitable for plants and animals on the damaged slope. Ecosystem function and ecological balance will also be restored over time [65,66]. The experiments of Jia et al. [67] showed that guar gum could increase the erosion resistance of fibrous loess, lead to its solidification by increasing the conservation of soil and water, and comprehensively enhance the ability of fibrous loess to maintain slope stability. In addition, ecological slope protection technologies such as wet sprayed concrete and hydraulic spray seeding have widely been used in engineering practice. A slope protection project between the borrow pit slope of the Mandela–Datong Highway (China) and the connecting canal of Huama Lake in Hubei (China) proved that wet spraying concrete technology could effectively stabilize slopes with a gradient of <1:0.75, increase their soil and water conservation capacity, and achieve slope revegetation in a short time [68,69]. Figures 1–3 show the effect of ecological restoration at the sandy slope of Gongga Airport in Tibet, the engineering slope of Jiacha Hydropower Station in Tibet, and the rock slope of Laohuzui Tunnel on Paimo Highway (China), respectively. By formulating a targeted ecological slope protection scheme and using hydraulic spray seeding technology, the three engineering slopes achieved the goals of surface soil solidification and restoring natural vegetation. Further examples of ecological restoration of engineering slopes are presented in Table 3. Some commonly used technologies for protecting ecological slope are summarized in Table 4 [3,11,12,70–90].

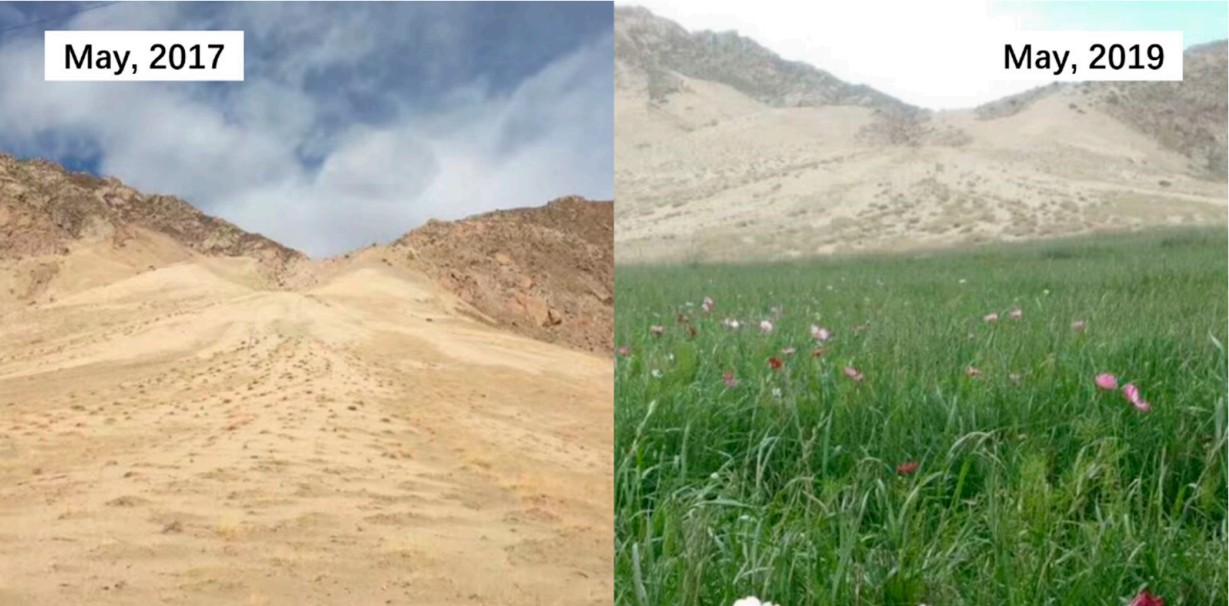

**Figure 1.** Comparison of restored and unrestored engineering slope at Gongga airport in Tibet.

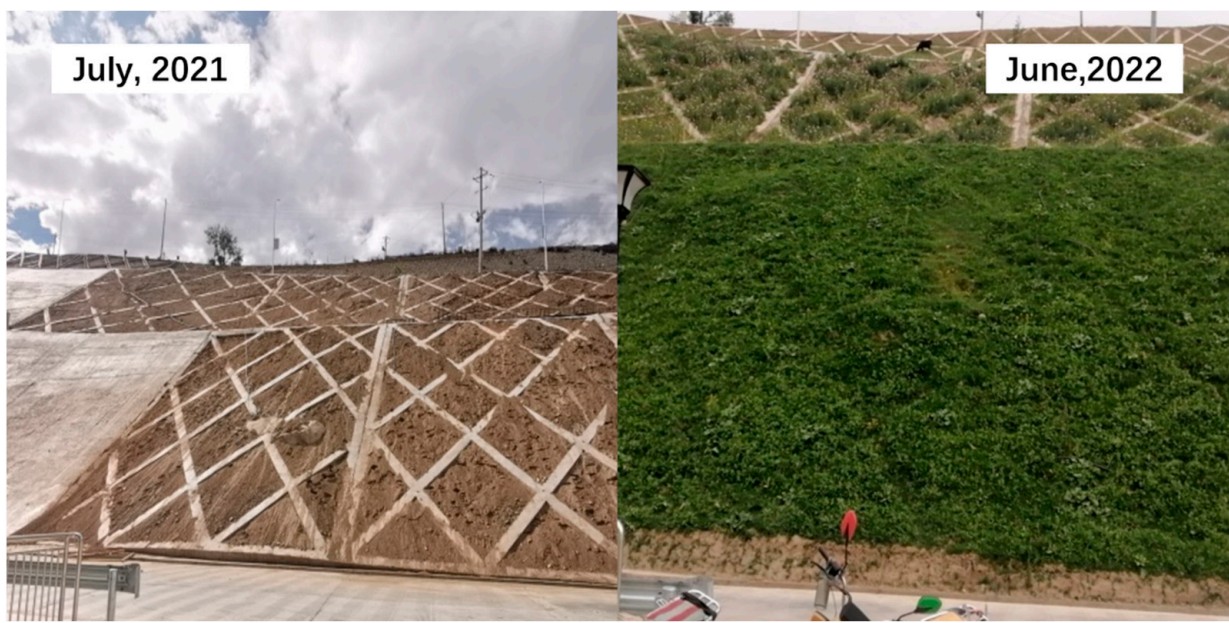

**Figure 2.** Comparison of restored and unrestored engineering slope at Jiacha Hydropower Station in Tibet.

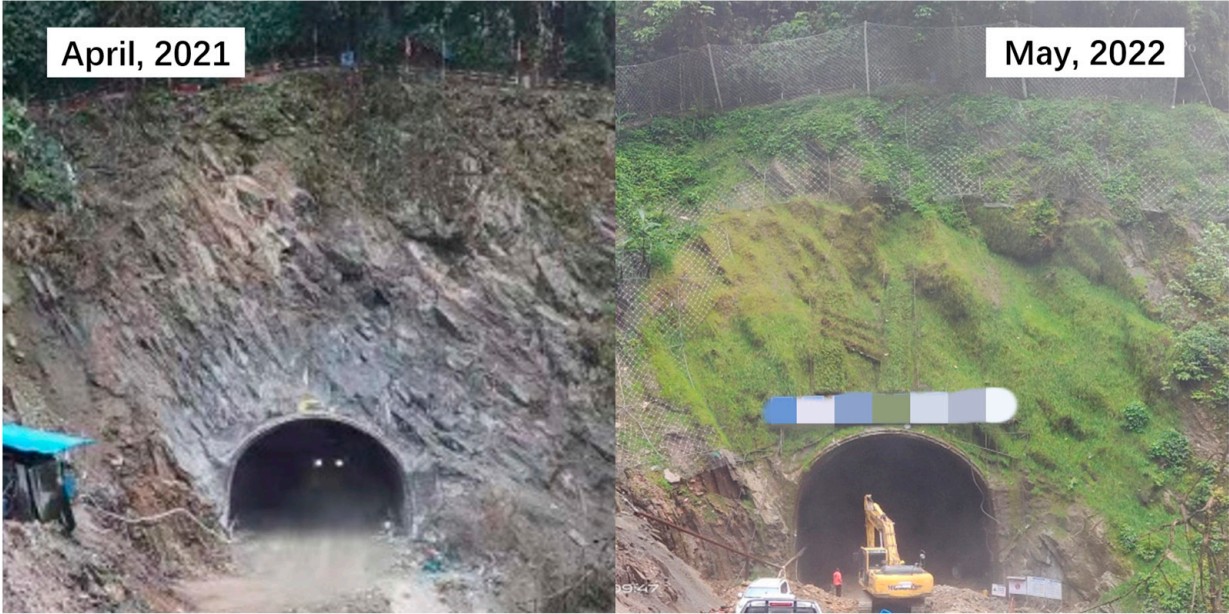

**Figure 3.** Comparison of restored and unrestored engineering slope at Laohuzui Tunnel entrance of Paimo Highway.

**Table 3.** Some examples of ecological restoration of engineering slopes.

| Challenges | Technologies | Results | References |
|---|---|---|---|
| The rocky slope is located in a cold region in northeast China, where the annual mean temperature is below 0 °C, making it difficult for plants to grow. | Thick layer planting and vegetation concrete | Both ecological restoration technologies built stable vegetation communities close to the natural ones. The impact of ecological restoration is better on the sunny slope than on the shady slope. | [91] |
| This rocky slope is located in a cold and high-altitude region on the northwest Sichuan plateau. The altitude is 3300 m, and the temperature varies from −3.8~3.3 °C. | Thick layer planting | The slope was well covered by vegetation. Further, nutrient contents in the soil were maintained at a high level during the two hydrological cycles after the ecological restoration. | [92] |
| The steep quarry slope is located the northwest of Ma'an Mountain, Hubei Province, with a maximum height of approximately 90 m and a slope gradient of 70~80°. | Soil spray-sowing | Half a year after the ecological restoration project, the vegetation coverage reached more than 95%. However, because of the impact of a rainstorm, the vegetation coverage decreased to less than 40%. | [93] |
| The rocky slope is located in a rainy region in southeast China, where it is seriously damaged by quarrying. Soil erosion and landslides are often triggered during rainfall. | Soil spray-sowing | The vegetation coverage and diversity were significantly increased. At the same time, the probability of soil erosion and landslides was significantly reduced because of soil fixation by the vegetation root system. | [94] |
| The loess slope is located on the south-central of the Loess plateau, where it is difficult to breed and propagate plants with intense evaporation and an arid climate. Moreover, loess flow slides and shallow landslide erosion occurred frequently in this region. | Planting | Screening and planting suitable species for ecological restoration and slope stability in the Loess plateau effectively increased vegetation coverage and prevented loess flow slides and shallow landslides. | [95] |

**Table 4.** Commonly Used Ecological Slope Protection Technologies for Engineering Slopes.

| Technology | Description | Advantages | Ranges of Application | References |
|---|---|---|---|---|
| Planting | Includes sowing grass seeds, vegetation cultivation, etc. | Simple construction and low cost | Slopes with small area and sandy slope | [72] |
| Planting stripe | Lay multi-layer non-woven geotextiles or natural fiber mats with plant seeds on the slope surface | Sowing and fertilizing are uniform, the quantity is accurate, the method is simple, UV-resistant, corrosion-resistant, non-degradable, and resistant to animal damage | Soil slope with gentle gradient or sandy soil slope | [12,72,74,88] |

**Table 4.** *Cont.*

| Technology | Description | Advantages | Ranges of Application | References |
|---|---|---|---|---|
| Soil spray-sowing | A mixture of green plant seeds, fertilizer, organic glue, water-retaining agents, grass fiber, colorant, etc., and water is evenly sprayed on the slope by hydraulic sprayer | The construction is simple and quick, has a wide range of application, and can effectively inhibit the erosion of the slope | Soil slope with gentle gradient or sandy soil slope, mixed soil slope | [71,72,75] |
| Mesh Bag Project | After taking engineering measures to consolidate the soil and stabilize the slope, lay the fiber mesh or metal mesh bag containing plant seeds, fertilizers and soil on the surface of the slope | Wide application range and strong adaptability | Soil and rocky slopes | [3,72] |
| Frame beam | After the concrete frame is built on the slope, fill in soil or soil bags in the interspace | Strong anti-scour ability, high construction efficiency, and low cost | Steep soil slopes and easily weathered rock slopes | [72,78] |
| External-soil spray seeding | Using a wet spray gun, spray the mixture of plant seeds, fertilizer, soil, and water on the slope surface to form a 1–3 cm thick vegetation layer, and then lay non-woven fabric or emulsified asphalt | Wide range of applications, high construction efficiency, high early vegetation coverage, good vegetation coverage speed and sustained effect, and good effect on controlling slope soil loss | Small and flat slopes and excavation slopes with few soil components and high hardness | [72,79] |
| Thick layer planting | After cleaning the slope surface and laying barbed wire, a mixture containing plant seeds, soil, organic matter substrate, water, binder, etc. is sprayed on the slope surface three times with a mortar spray gun. After spraying, cover with non-woven fabric and spray regularly for maintenance | Easy to construct, fast, safe, and reliable, and can form a stable, high fertility, and strong erosion resistance foundation | All kinds of rocky slopes | [70,72,80] |
| Planting after spraying OH liquid | Use special machinery to spray water-diluted OH liquid containing grass seeds on the slope surface to consolidate the topsoil of the slope and form an elastic solid film | Simple and fast in construction, does not require post-maintenance, and has good slope protection and greening effects | Barren soil slopes and heavily weathered rock slopes | [72] |
| Porous concrete for plants-growing | Fill the pores of the porous concrete with a mixture of seeds, water-retaining agents, organic fertilizers and fertilizers, so that the roots of the seeds pass through the pores of the concrete to the lower soil | Stable slope structure, beautiful appearance and regularity, and has good environmental benefits | Weathered rock, hard rock slopes, soil slopes or flats | [72,81] |
| Three-dimensional vegetation net | After planting designated types of turf on the soil surface, a geonet pad is laid, and through the growth activities of plants, the roots are reinforced and the stems and leaves are prevented from being eroded | Strong anti-scour performance, and has good engineering effect and social and economic benefits | Natural soil slopes, road and railway engineering slopes, water conservancy projects, etc. | [83,84] |

**Table 4.** *Cont.*

| Technology | Description | Advantages | Ranges of Application | References |
|---|---|---|---|---|
| Vegetation Concrete | The loam is mixed with Portland cement, vegetation concrete ecological modifier, organic fertilizer, and plant seeds, and sprayed onto the slope surface with dry and wet sprayers to form a semi-rigid slope protection structure | Good strength and long-term stability, strong erosion resistance, and good frost resistance and sustainability | Applicable to all kinds of earth-rock slopes, hardened slopes, rocky slopes; the limit using slope ratio $\leq$ 1:0.1 | [76,85] |
| Hybrid Fiber Spraying Technology | Incorporates high-order agglomerates and soil conditioners into the foreign soil material rich in organic matter and clay, and then sprays through a seeder to generate agglomerate reactions to form artificial green growth substrates similar to nature | A wide range of applications, good repair effect, and convenient construction | Mostly used for rock slopes with a slope ratio in the range of 1:0.2–1:1 | [11,86] |
| Net-suspended spray seeding | After leveling the site, lay diamond-shaped wire mesh from top to bottom, and spray the substrate and soil containing shrub and grass seeds | The hanging net can make the substrate and the soil form a whole, improve the water retention of the soil, and has a strong anti-scour ability | Steep slope | [73,87,89,90] |
| Re-greening Technology of Grading and Retaining Platform | Combined with engineering measures such as retaining walls and planting troughs, the high and steep slopes are grading and retaining platforms, and climbing native plants with strong resistance are planted in the planting troughs to solve the problem of vertical greening | Makes full use of the remaining resources with less investment | Abandoned open pit mine with steep slopes | [77] |

## 4. Potential of Benefits and Evaluation of the Impact of Ecological Restoration of Engineering Slopes

### 4.1. Benefits of Ecological Restoration of Engineering Slopes

The benefits of ecological restoration mainly relate to the sustainability of the vegetation community, physical and chemical properties of soil and nutritional status, improvement in landscape pattern, and benefits from the conservation of soil and water [62]. The benefits of ecological restoration can be evaluated by comparing the soil water content, vegetation coverage, and other related indicators of different states of restoration. In a mountain-oasis-desert system in an arid area, an ecological restoration project clearly increased the vegetation coverage, soil water content, and net primary productivity over time. Additionally, the increased soil conservation and biodiversity status evidenced the benefits of ecological restoration [96]. Moreover, slope ecological restoration on the west slope of Erlang Mountain (China) resulted in a rapid increase in the coverage of herbaceous plants, followed by a slow decrease in the long-term ecological restoration. Different from herbaceous plants, the coverage of shrubs showed a constant upward trend [97]. These reports showed evidence of a local succession, from herbaceous-dominated to a shrub-grass-dominated community, concomitantly with a steady increase in the ecological protection function and sustainability of vegetation.

By comparing the restored and the original ecosystems, the benefits of ecological restoration can also be evaluated. A study on the benefits of the ecological restoration of Yongxing Coal Mine in Zichang County, Shanxi, China, showed that the restoration project improved the ecological environment of the construction area, with the damaged area starting to perform its original functions, such as the conservation of soil and water [98].

### 4.2. Evaluation of the Ecological Restoration Effect

At present, the evaluation of the effect of ecological restoration is mainly conducted through single-factor indicators or the multi-indicators system [96]. Single-factor indicators are vegetation-related, such as plant greenness, coverage, and primary productivity, and are often used to judge whether the main purpose of ecological restoration has been accomplished or not. Despite being able to comprehensively evaluate the effects of ecological restoration, a multi-indicator system is difficult to achieve in the actual evaluation process. Therefore, diversity of species, vegetation structure, and ecological processes are often used as substitutes of multi-indicator system in engineering practice [96]. For example, Anna L. Dietrich et al. [99] proposed an evaluation system of ecological restoration with the use of the growth state of plants or specific organs as an indicator to estimate whether the ecological restoration is effective or not.

As an important part of an ecological restoration practice, the evaluation of its effect provides a standard, which is used to determine whether the ecological restoration is effective or not. It is also the basis for the formulation of an ecological restoration plan and for managing the ecosystem [100]. Various indicators for evaluating the effect of ecological restoration of an engineering slope are presented in Figure 4 [101].

Remote sensing, with the advantages of short revisit time, large monitoring range, easy and rapid access, and high spatial resolution, is widely used for evaluating the effect of the ecological restoration of engineering slopes. Compared to field survey, the use of remote sensing technology in evaluating the effect of the ecological restoration of engineering slopes leads to more accurate and reliable results while spending less time and cost [102,103]. Zeke Lian et al. [104] analyzed and evaluated the dynamic changes between the time period of 2014~2021 in the ecological and environmental quality of tertiary relic reserves with different degrees of development and protection in the Tongluo Mountain Mining Park in Chongqing, China by using Landsat remote sensing images as the data source and the four remote sensing ecological indices of vegetation, heat, humidity, and dryness as the evaluation index. Jiawei Hui et al. [105] proposed a normalized environmental vegetation index, which was based on the Landsat and Sentinel data and selected eight remoted

sensing data within the time range of 2005~2019 to reflect the changes in vegetation after ecological restoration of a grassland mining area in Xilinhot City, China. It can be seen from the above that remote sensing technology plays an important role in the evaluation of the effect of ecological restoration of engineering slopes. However, it is not advisable to use remote sensing separately to evaluate the effect of the ecological restoration of engineering slopes. This is because some of the evaluation indicators, such as public satisfaction, cannot be calculated using remote sensing. Synthesizing the indicators shown in Figure 4—which were calculated using the data obtained through field survey and remote sensing technology within the evaluation period and evaluation base period—is a wise choice to lessen the bias of the evaluation of the effect of engineering slopes.

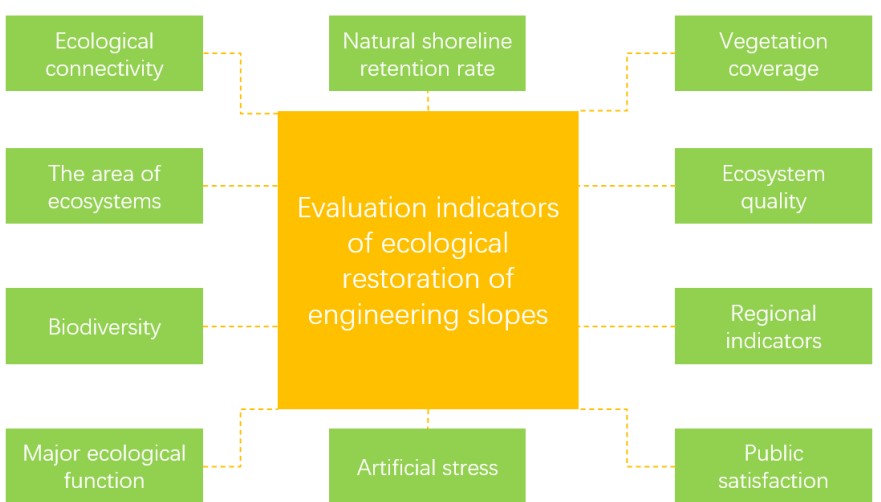

**Figure 4.** Ten evaluation indicators of the ecological restoration of engineering slopes.

## 5. Challenges and Drawbacks of Ecological Restoration of Engineering Slopes

With the continuous practice and research on the ecological restoration of engineering slopes, China has made great progress in the field of ecological restoration. For example, the method of ecological slope protection, which combines soil improvement and bioremediation is widely accepted. Further, the process of ecological restoration is now highly mechanized. Even so, there are four problems that still need to be addressed:

(1) In some projects, the plan of ecological restoration is not formulated in accordance with the local geographical conditions. In the vast territory of China, each local ecological restoration project faces different problems. Even so, sometimes the local conditions have not been taken into consideration in some projects, to reduce costs or for other reasons. It is necessary to scientifically classify different areas, set reasonable local ecological restoration goals, formulate targeted restoration plans, and build an effective evaluation system to study the effect of restoration throughout the different areas.

(2) Long-term ecological and environmental protection of the restored engineering slopes is sometimes ignored in the construction industry in order to gain more revenue. For example, after restoring the engineering slope through vegetation concrete, it is difficult to form a stable biological community on the restored engineering slope by ignoring the maintenance of vegetation or using inappropriate maintenance methods to reduce costs [76]. A large number of new engineering slopes are inevitable as the demand for infrastructure in China remains high. However, long-term ecological protection should always prevail over short-term benefits. An equilibrium between long-term ecological benefits and short-term economic needs has to be reached before the construction of project, selecting appropriate ecological protection and restoration methods, and carrying out timely ecological restoration during or after the construction.

(3) Due to the lack of quantitative evaluation of the ecological restoration of engineering slopes, the evaluation can be ambiguous at times. Nowadays, an ecological restoration

process is still assessed considering whether there was an improvement in the quality of the environmental or not, and evaluated through qualitative means such as "excellent" or "good" [106]. This leads to an evaluation bias, a result of subjective feelings of the evaluators, which eventually leads to different evaluations of the same project. This also reduces the credibility of the evaluation, ultimately affecting its guiding role in the restoration practice. Scientific and standardized quantitative conclusions are still lacking in the evaluation of the ecological restoration of engineering slopes.

(4) Long-term monitoring of restored slopes is sometimes lacking. As most scholars agree nowadays, long-term monitoring of the dynamic process of ecological restoration is necessary [107]. However, in practice, the evaluation of the effect of ecological restoration is mostly based on shorter period, ignoring the long-term nature of ecological restoration and the dynamic change of vegetation.

Accordingly, some prospects for the ecological restoration of engineering slopes in China are proposed. Firstly, the quality and efficiency of restoration will be improved by using more advanced restoration technologies or methods. As the study of ecological restoration continues, more environment-friendly technologies will be invented, coupled with more advanced methods that could come into the picture due to technological advancement, thus greatly improving the quality and efficiency of ecological restoration. Secondly, more attention will be paid to the long-term monitoring of restored slopes. The long-term monitoring of restored slopes, which is agreed upon by most scholars, is sometimes ignored in the construction industry. With the perfection of relevant regulations or laws, this problem will be solved in the near future. Thirdly, a more accurate evaluation system that is recognized by most scholars will be proposed. The qualitative evaluation system adopted today does not meet the requirements of accuracy in the evaluation of the effect of ecological restoration. A commonly recognized evaluation system, which is standardized and quantitative in nature, may be established through closer international exchanges.

## 6. Conclusions

With the development of China's economy and social productivity, there is an increase in construction projects, resulting in an increase in the amount of engineering slopes. The proposal of environmental protection concepts such as "lucid waters and lush mountains are invaluable assets" shows that ecological environmental protection has become an indispensable part of the social and economic development of China. However, it is not enough to just put forward some concepts of environmental protection. There is an urgent need to build a standardized system of ecological restoration of engineering slopes; that is, to formulate a suitable restoration plan for each ecological restoration project, select an appropriate restoration technology, and perform long-term maintenance and monitoring after the ecological restoration until a stable ecological community is formed. Further, a more rigorous, scientific, and accurate evaluation system of ecological restoration is also needed. Synthesizing the evaluation indicators of the ecological restoration of engineering slopes, calculated by the data obtained through field survey and remote sensing technology, may be the preferred strategy. By summarizing the existing ecological restoration technologies and evaluation systems, this paper addresses the progress and provides valuable information for future projects and evaluations, which will contribute to the long-term sustainability and rapid development of the ecological restoration of engineering slopes in China.

**Author Contributions:** The whole article is the result of shared effort. Conceptualization, Y.S. and Q.L.; methodology, Y.S. and Q.L.; software, X.P.; validation, Q.L. and X.P.; investigation, Y.S., R.W. and B.Y.; writing-original draft preparation, Y.S., Q.L., R.W. and B.Y.; writing—review and editing, Y.S. and Q.L.; supervision, N.L. and X.Z.; resources, D.Y. and S.W.; project administration, Q.T. All authors have read and agreed to the published version of the manuscript.

**Funding:** This research was funded by [Chengdu University of Technology Research Startup Fund] grant number [10912-KYQD2021-01944], [Open Fund of State Key Laboratory of Geohazard Prevention and Geoenvironment Protection] grant number [SKLGP2022K023, SKLGP2021Z017 and SKLGP2021Z018], [Project of 'Study on Influence Factors and Key Technologies for Survival of the Replacement of Stripped Sod in Alpine Meadow'] grant number [AH2022-0295], and [Project of 'Ecological Restoration of Damaged Slopes in the Middle Reaches of Yajiang Power Station Technology System Research and Application'] grant number [JC2020/D02]. And the APC was funded by [Project of 'Study on Influence Factors and Key Technologies for Survival of the Replacement of Stripped Sod in Alpine Meadow'].

**Institutional Review Board Statement:** Not applicable.

**Informed Consent Statement:** Not applicable.

**Data Availability Statement:** There were no new data in this manuscript.

**Conflicts of Interest:** The authors declare no conflict of interest.

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
