# Peer review of "Ecological Restoration of Engineering Slopes in China—A Review"

_sustainability, doi:10.3390/su15065354_

Round 1
Reviewer 1 Report
Taking account that this article is a literature review of the topic "Ecological Restoration of Engineering Slopes", from my point of view 2 very important aspects are missing:
- Describing a search methodology and selection of articles worldwide in the last 25 years, dealing with the main articles that have an impact on the subject. In this respect, it would be important to take account the following for each of the articles cited:
§ Subject, group and position of the journal according to the impact factor at the time the article in question was published.
§ Number of citations of each article.
§ Ranking of the citations of each article in the 4 quartiles in which the journals are classified according to the impact factor.
- Illustrate each methodology of the subject "Ecological Restoration of Engineering Slopes", with some of the most important examples observed in the different articles.
Reviewer 2 Report
1 - Figures and tables
The 4 figures and 2 tables referred to in the text were not sent to the reviewer. Therefore, there is no condition to carry out its full analysis.
2 - Text
The text is intended to provide an implementation and evaluation reference for the ecological restoration of engineered slopes in China, but has no technical content to do so, containing too many common knowledge statements
Although the text cites many scientific papers, it only touches on the subject very superficially, without providing strong scientific considerations or justifications for the technical aspects of ecological restoration of engineered slopes.
Reviewer 4 Report
The study is interesting and fits within the scope of Sustainability. The finding will be of high interest to the engineering, ecologic community, and broader scientific community. However, substantial modifications should be made by the authors before the publication.
The tables are not found in the manuscript!
It would be good to specify the term “engineering slopes” in the abstract.
“effect evaluation” should be “impact evaluation”
Double check the required format of the citations and references. Such as “States (Yu, et al. 2010)”,
Discussion: Please present a critical discussion, not just a descriptive summary of the topic.
Is there any application of ecological restoration to highway and tunnel slopes in cold regions, such as “asymmetric talik formation beneath the embankment of Qinghai-Tibet highway triggered by the sunny-shady effect”, “Impacts of snow cover on the pattern and strength of air flow in air convection embankment in sub-arctic regions”, “spatiotemporal vegetation cover variations associated with climate change and ecological restoration in the loess plateau”
Round 2
Reviewer 2 Report
I consider that the authors have made the improvements indicated by the reviewers and have greatly increased the quality of the work. A general review of technical terms in English should be made.
Reviewer 4 Report
Accept
Author Response
Dear Reviewer 4:
We appreciate your precious time reviewing our manuscript (manuscript ID: sustainability-2200550) and your affirmation of our manuscript.